# Monitoring for Coastal Resilience: Preliminary Data from Five Italian Sandy Beaches [note 1]

**DOI:** 10.3390/s19081854

**Published:** 2019-04-18

**Authors:** Luca Parlagreco, Lorenzo Melito, Saverio Devoti, Eleonora Perugini, Luciano Soldini, Gianluca Zitti, Maurizio Brocchini

**Affiliations:** 1Istituto Superiore per la Protezione e la Ricerca Ambientale, 00144 Rome, Italy; saverio.devoti@isprambiente.it; 2Department of DICEA, Università Politecnica delle Marche, Ancona 60131, Italy; l.melito@pm.univpm.it (L.M.); e.perugini@pm.univpm.it (E.P.); l.soldini@univpm.it (L.S.); g.zitti@univpm.it (G.Z.); m.brocchini@univpm.it (M.B.)

**Keywords:** video-monitoring, beach morphological evolution, beach resilience, sand bars

## Abstract

Video-monitoring can be exploited as a valuable tool to acquire continuous, high-quality information on the evolution of beach morphology at a low cost and, on such basis, perform beach resilience analyses. This manuscript presents preliminary results of an ongoing, long-term monitoring programme of five sandy Italian beaches along the Adriatic and Tyrrhenian sea. The project aims at analyzing nearshore morphologic variabilities on a time period of several years, to link them to resilience indicators. The observations indicate that most of the beach width variations can be linked to discrete variations of sandbar systems, and most of all to an offshore migration and decay of the outermost bars. Further, the largest net shoreline displacements across the observation period are experienced by beaches with a clear NOM (Net Offshore Migration)-type evolution of the seabed.

## 1. Introduction

A natural system is resilient when it is able to restore perturbations without losing its functionality. In the same way, the resilience of a beach could be defined as the capability to preserve its functionality under changing hydro-morphological conditions. Among the main and most relevant functions of wave-dominated beach systems is its capability to dissipate and absorb the energy of waves coming from the offshore.

The morphologic impact of waves on coasts, focus of a growing volume of literature (e.g., [1,2,3]), has been proven a crucial matter for the subsistence of coastal communities. The thrive of coastal ecosystems and the related benefits for society are closely linked to beach integrity and well-functioning. Thus, efficient management policies require evidence-based strategies to protect coasts without affecting the beach system capability to adapt to the hazards of climate change [4]. A clear example of beach resilience are the fluctuations in shoreline position in response to a storm [5]. The shoreline recovery acts as a long-term process of maintenance of the subaerial beach width and affects the net impact of storms in time [6,7,8,9,10].

Nearshore sand bars are a crucial component of sandy coastal systems, since they act as sediment storage and wave energy preferential dissipation points, thus playing a key role in the overall morphologic stability of the shoreline. Generally, during large wave events sand is shifted offshore and deposited in submerged sandbars [11]. During the following calm periods, sandbars slowly migrate back onshore and possibly weld to the shoreline to restore the pre-storm subaerial beach width and morphology [12].

Relationships between the shoreline and the sandbar morphology were first described by the “morphodynamic model” of Wright and Short [13], in which beaches were classified into specific morphological states, each characterized by different sandbar configurations and connections to the shoreline. Transitions between morphological states are described in the model as a response to wave and tide forcings [14].

Among the most commonly recognized sandbar evolutionary processes is the Net Offshore Migration (NOM) pattern [15], characterized by a long-term cyclic evolution in three stages [16]. At first, a bar is generated close to the shoreline by interaction of undertow, wave velocity asymmetries and long waves effects [17]. The bar then experiences a net offshore migration, a byproduct of an alternation of gradual onshore movements during calms and strong offshore motions during storms [18]. Finally, the bar moves offshore of the breaking line and degenerates [19]. NOM cycles have been identified on several barred beaches worldwide. In the Mediterranean Sea this phenomenon was observed along the Gulf of Lions and described both by bathymetric [20] and remote-sensed data [21]. Relations between the extent and typology of beach erosional phenomena and bar stage are observed [15] since a decaying outer bar offers a reduced protection to the shoreline and inner bar, leading to enhanced erosion of the shoreline [22,23,24]. Recently, the analysis of shoreline recovery processes has provided further insight in the close relation between shoreline recovery rates, the bar location and the tidal regime [10,25].

In the past decades, the increase of remote sensing techniques for the monitoring of coasts have promoted a deeper understanding of the multi-year response of the beach system, allowing for a much greater spatial and temporal resolution than traditional survey methods. Moreover, they ensure that all significant events involving coastline alteration can be analyzed in a retrospective way [26]. In particular, the evaluation of barlines and shoreline positioning, as well as their medium-to-long term evolution, via remotely sensed images has been proven a rather affordable and reliable approach that allows for the collection of a large volume of data. Several metrics for a thorough analysis of beach response to hydrodynamic forcing can then be extracted by video-monitoring products. Among these are the wave run-up [27,28] which can be used as a proxy for the residual wave energy reaching the shore after wave breaking, the swash zone properties (e.g., see [29]), and the mean shoreline net changes after storm events (e.g., see [30]). A number of video-monitoring stations have been productively deployed along Italian coasts in recent years (e.g., see [30,31,32,33,34]) and their informational value can be proficiently used to satisfy the increasing need for informed coastal management decision planning by local authorities [35].

In this paper, we present preliminary results of nearshore morphological evolution derived from multi-year beach monitoring. The study sites are located along five Italian beaches characterized by different coastal protection demands and environmental forcing. This work is part of an ongoing, long-term monitoring programme of sandy beaches belonging to environmentally-protected areas, with the aim of providing the information on beach resilience needed to ensure the survival and good functioning of such environmental sanctuaries [36].

Section 2 provides an overview of the study sites, the local climates and video-monitoring system used for the analyses. The main results are detailed in Section 3. A discussion of the results and some concluding remarks are finally proposed in Section 4.

## 2. Materials and Methods

### 2.1. Study Sites

The study sites are located in central Italy, three along the Adriatic Sea and two along the central Tyrrhenian Sea (Figure 1). The sites present sandy beaches free from coastal defense structures and characterized by different wave exposures and seabed slopes. All the sites are micro-tidal beaches with the major tidal excursions experienced by the Adriatic beaches, in all cases the tidal excursion never exceeding 0.6 m (Rete Mareografica Nazionale—ISPRA). The main differences between the Adriatic and Tyrrhenian sites are due to inherited geological and physiographic constraints, resulting in different local wave climates, sedimentary input densities and degrees of coastal fragmentation by geological forcing (rocky headlands), as well as different amplitudes of storm surges and wave set-up, both larger in the semi-enclosed Adriatic basin.

The Senigallia beach is located along the northern portion of the Central Adriatic Sea, 1 km south of the Misa river estuary, in a region of touristic attractiveness that is included in the longest stretch of unprotected beach of the Marche region [37]. The coastline is oriented approximately NW–SE and the beach is characterized by fine and medium sands ($d_{50}\approx$ 0.25 mm), with a nearshore slope of about 0.5% computed, as for all the sites, as the average slope between the shoreline and the −10 m depth isoline.

The second Adriatic site is located along the Central Adriatic margin inside the marine protected area of Torre del Cerrano. The shoreline is NW–SE oriented and the beach is composed of fine/medium sand ($d_{50}\approx$ 0.3 mm) with nearshore slope of about 0.5–0.6%.

The Rodi Garganico site, part of the Gargano National Park, is the latest observing systems installed and is located along the northern side of the Gargano promontory in the southern Adriatic Sea. The coastline orientation is about E–W facing the north, with medium grained beach sediments ($d_{50}\approx$ 0.3 mm) and nearshore slope of about 0.8%. The harbour located 1 km eastward of the monitored beach may have an important influence on the morphological evolution of the nearby coastal areas, since it may alter the longshore sediment flow patterns significantly.

Two adjacent beaches were analyzed along the central Tyrrhenian side: Sabaudia and Terracina. They are separated by the rocky Circeo Cape and characterized by very different touristic demand. The Sabaudia sandy beach-dune system, spanning for 24 km northward of this rocky headland, is entirely included in the Circeo National Park since 1934, representing one of the largest and most natural littoral zones of the Tyrrhenian coast. The video-monitored portion is located about 5 km north of the Circeo Cape and is composed of fine to medium sand ($d_{50}\approx$ 0.3 mm) with a nearshore slope of about 1.3%. The site orientation is NNW–SSE.

The Terracina beach, located eastward of the Circeo Cape, is enclosed into a 15 km long embayment with an average WSW–ENE orientation and a nearshore slope of about 1.7%, the maximum nearshore slope value of all the five sites here studied. Two major rivers are present inside the embayment and the analyzed sector is bounded at one end by one of these river jetties. Native beach sediments consist of fine to medium-fine grained sand ($d_{50}\approx$ 0.3 mm) evolved into mixed beach sediments after a beach face nourishment was deployed in 2007, with a mean beach infill of 270 m^3^/m [31]. Because of this, the evolution of the Terracina beach has to be regarded as a long-term self-adjustment of a “not in equilibrium” beach. This is a situation different from those of the other sites, where no relevant coastal interventions were undertaken before and during the monitoring period, and thus the natural beach adaptation processes in response to local wave forcing is not artificially altered.

### 2.2. The Video-Monitoring System and Data Processing

All the analyzed areas are provided with double-camera video-monitoring stations working with the same setting since 2016, except the Rodi Garganico site, operative since the end of 2018 and, for this reason only, partly described in this manuscript. The Terracina system, previously made of a single-camera system, was upgraded to the actual double-camera setting in 2015.

The monitoring stations are all located on the roof of beach-front buildings, at a height larger than 20 m above the mean sea level (Table 1) and at a distance smaller than 150 m from the shoreline. The acquiring systems routinely collect images of the nearshore zone at 2 Hz during 10 min every daylight hour, through a double digital video-camera equipped with CMOS 1/1.8${}^{\prime \prime }$ sensor and fitted with fixed-focal lens. The alongshore extent of the monitored coast is site-specific, but always ensures visibility of the active nearshore zone along a sector extending 500–1000 m in the longshore direction. Information on camera systems are given in Table 1.

In order to achieve quantitative morphologic information from remote images, standard photogrammetric procedures [38] were used to project the oblique video image onto a planar surface, coincident with the local sea level [39]. The geometrical solutions of the projection procedure were constrained with several Ground Control Points (GCP), clearly visible points in the image with known coordinates both in the image-space and in ground-space. The geometries were computed for each selected image, corrected for optical distortions following the method used by Bouguet [40] and finally transformed into geometrically-correct plan views on a 1 m grid. A local metric reference system was set for each station, with the origin fixed at the video-station position and rotated in order to ensure that the mean shoreline direction is parallel to the *x*-axis and the *y*-axis is positive toward the offshore (Figure 2). The cross-shore and alongshore limits of the analyzed portion of each rectified image were fixed at distances from the stations where the pixel footprint was not higher than 5 m and 20 m, respectively.

Quantitative information on the morphological features of the nearshore zone has been extracted from optical signatures visible on the optical images and used to investigate the dynamics of the analyzed area [39,41]. Shoreline and sandbar locations have been indirectly estimated using composite images resulting from time averaging of the pixel intensity (Timex images). In this way moving features, e.g., propagating waves and bores, are not visible but more stable characteristics, e.g., regions of wave breaking, are highlighted as white bands [42].

Once Timex images have been rectified and georeferenced, many approaches to sample patterns of high intensity from plan view images can be used. In the present work the Barline Intensity Mapper (BLIM) by van Enckevort and Ruessink [18] and Pape et al. [43] has been used. In this algorithm, pixels within a region of interest are scanned along the image in order to detect the position lines of maximum pixel intensity values (Figure 2). These lines have been taken as proxies for wave breaking regions and, therefore, for the positions of sandbar crests and shorelines with a suitable accuracy.

One single image per day was chosen close to the lowest tide level, when environmental conditions allowed to clearly analyze the image. For each sampled feature (bar line or shoreline), the cross-shore position was alongshore averaged and used as a proxy for the overall cross-shore position of the remotely sensed morphology. Alongshore standard deviation could also be assumed as representative for the alongshore unevenness of the morphology. Other morphological parameters, like the bar height and the existing bathymetry, are also deemed to be relevant to the beach morphodynamic processes; unfortunately, no information on this regard could be extracted by video-monitoring products in our study, although there are some undergoing studies on the matter of estimating the bathymetry from remotely sensed images [44].

Since the location of barlines and shorelines could be affected by uncertainties related to sea level existing during the Timex construction [18,45], a conservative approach was used to filter out sea level-induced errors on features positions. Cut-off resolutions were fixed at ±5 m for the shoreline, ±10 m for the inner bars and ±20 m for the outermost bar, so that all consecutive feature motions lower than such thresholds were not regarded as real cross-shore displacements. The present analyses focused on the long-term behaviour rather than on the single event response, allowing, however, to identify all beach feature variations whose amplitude were larger than the above-mentioned thresholds.

Further, in order to analyze the long-term behaviour of selected beaches, a down-sampling of time series was operated by averaging values over different temporal windows (7 days, 15 days and 30 days). Some periods, however, remained unsampled because of system failure, dirty lenses due to persisting bad weather, or calm weather giving no visible wave breaking patterns.

Throughout the manuscript the shoreline is labelled as “sl” and bars as “b*n*”, where *n* is an identifying number that increases with the temporal occurrence and the offshore position of the given feature. Only for the beach in Sabaudia, the presence of transient sandy features between the bars and the shoreline (labelled as SPAW, Shoreward Propagating Accretionary Waves [46]) has been detected.

Table 2 gives basic information on the time extension and a summary of the sampled features in each image dataset, with the number of days each feature appears, and its percentage of occurrence over the whole observation period.

### 2.3. Wave Climate

2006–2016 hindcast and 2017–2018 forecast wave data delivered by the Copernicus Marine Environment Monitoring Service (EU Copernicus programme, http://marine.copernicus.eu/) have been used to characterize wave forcings typical of the study sites. For the mid/long term analysis described here, a preliminary wave height threshold equivalent to a 97th percentile has been adopted [5,47]. In other words, the threshold for the definition of a storm event is assumed so that the recorded time series of wave heights stays above it for 3% of the whole analyzed period. The proposed method identified about 150 events per region as classified storm events throughout the period 2006–2018. As suggested by Boccotti [48], two consecutive storm records must be regarded as distinct if they are separated by a continuous interval greater than 12 h. The wave data and characteristics are here primarily used to compute morphodynamic parameters (Section 3.1), and no specific analysis was performed to characterize storm statistics. Statistical information on the storm wave forcings at the investigated sites are summarized in Table 3.

Since the variability of nearshore patterns is highly related to the balance between cross- and alongshore components of the wave-induced currents, a crucial parameter to be considered across all sites is the wave approach angle with respect to the coast. As a consequence of the physiographic differences between the Adriatic and Tyrrhenian basins (elongated and enclosed the former, more open the latter), wave obliquity in the Adriatic sites is clearly bimodal (due to the action of the local Scirocco and Bora winds), while the incidence distribution for Tyrrhenian sites is mainly unimodal (Figure 3).

Along the Adriatic sites, the prevailing NE and E waves approach the coasts at different angles: at Senigallia the waves coming from N approach the coast at very large angles with respect to the shore normal, whereas at Torre del Cerrano waves approach the beach from two symmetric directions with large angles. At Rodi Garganico waves coming mainly from N–NW approach the coast almost normally (Figure 3; the panel pertaining to Rodi Garganico is not shown).

Concerning the Tyrrhenian sites, although both Sabaudia and Terracina are subjected to roughly the same wave climate, the dominant waves coming from W, SW and S induce different inshore hydrodynamics since the respective coastlines have strongly different orientation. This results in a dominance of waves approaching at low angles for Sabaudia, and at high angles for Terracina.

### 2.4. Beach Classification Parameters

A simple and general description of beach morphology at the investigated sites can be achieved by means of synthetic parameters, such as the dimensionless fall velocity Ω [13], the bar parameter $B^{*}$ [49], the relative tide range RTR [14] and the Iribarren parameter Ir [50]:(1)$$\Omega =\frac{H_{b}}{w_{s}T_{p}},$$
(2)$$B^{*}=\frac{x_{s}}{gT_{p}^{2}\tan\beta },$$
(3)$$\mathrm{RTR}=\frac{\mathrm{TR}}{H_{b}},$$
(4)$$\mathrm{Ir}=\frac{\tan\beta }{\sqrt{H_{b}/L_{0}}},$$
where $H_{b}$ is the breaking wave height, $w_{s}$ is the settling sediment velocity, $T_{p}$ is the peak wave period, $L_{0}$ is the estimated offshore mean wave length, $x_{s}$ is an offshore distance corresponding to a specific depth at which the beach slope becomes very small (beach “closure”), *g* is the gravitational acceleration, tanβ is the bottom slope and TR is the tide range. The settling sediment velocity is evaluated with the Zanke formulation [51,52], valid for sediment diameters in the range 0.1–1 mm:(5)$$w_{s}=10\frac{\nu }{d_{50}}\left(\sqrt{1+\frac{0.01gd_{50}^{3}\left(\rho_{s}-\rho_{w}\right)}{\nu^{2}\rho_{w}}}-1\right),$$
where ν is the kinematic viscosity of water, $d_{50}$ is the sediment median-diameter, $\rho_{s}$ is the particle density, $\rho_{w}$ is the water density.

The breaking wave height $H_{b}$ appearing in Equations (Equation 1) and (Equation 3) has been computed starting from the mean value of the (offshore) peak wave heights of the identified storms, $\langle H_{s,peak}\rangle$. The offshore datum has been transported to the breaking line by taking into account the effect of wave shoaling over an equilibrium beach profile:(6)$$H_{b}=\langle H_{s,peak}\rangle \;k_{s}$$
where $k_{s}$ is the shoaling coefficient. The wave period $T_{p}$ used in Equations (Equation 1) and (Equation 2) has been calculated as the mean of the peak wave periods at the peaks of identified storms $\langle T_{p,peak}\rangle$, and the Iribarren parameter as the mean of the Iribarren numbers computed for each classified storm event.

## 3. Results

### 3.1. Beach Classification

The classification parameters estimated for each site are summarized in Table 4. The dimensionless fall velocity Ω takes into account both wave and sediment characteristics; values of Ω larger than 6 have been found for all sites, allowing us to classify all the beaches as dissipative. In this kind of environment submerged bars may be present and rips are usually absent [13,14]. Low values of the surf similarity parameter, Ir (see Table 4), also display a dissipative beach behaviour, characterized by spilling type breaking waves (Ir <0.5). Following the conceptual beach model of Masselink and Short [14] that makes use of the relative tide range RTR, all beaches can be classified as barred dissipative beaches, further confirming the presence of submerged bars.

To predict the possible number of sandbars, use is made of the dimensionless bar parameter $B^{*}$ [49], which considers both the nearshore geometry and the wave characteristics (Table 4). According to the bar parameter, the presence of three or more bars has been estimated for Senigallia, Torre del Cerrano and Rodi Garganico ($B^{*}>100$), while only two bars are predicted for Terracina and Sabaudia ($50<B^{*}<100$). As it is shown in the following, these estimations are in fairly good agreement with the bar systems actually observed from the deployed video-monitoring stations (see Figure 4).

The method by Wright and Short [13], based on the parameter Ω and developed for energetic wave climates, usually does not provide a correct estimation of the morphodynamic states for enclosed basins (e.g., [53,54]). Nonetheless, this classification method still identifies all the analyzed beaches as dissipative, similarly to the findings of Lisi et al. [55] for the south Adriatic coast.

The above shows that, for inter-site comparison purposes, environmental parameters like the wave obliquity (Figure 3) and the nearshore slope (Table 4) are both important, as they significantly affect the nearshore hydrodynamics and the way in which sites are classified.

### 3.2. Nearshore Morphological Settings in Time

The seabed morphologies of the study sites are characterized by a number of nearshore bars variable from one to four. Only in Sabaudia the number of bars remained constant throughout the whole observation period (Figure 4 and Table 5).

The morphological setting in Senigallia (evaluated in an observational period spanning 2.1 years) is characterized by a 4-3-4 bars transition, with the 4-bar setting persisting for about 60% of the total observation time. During the 2.7 years of observations at Torre del Cerrano the nearshore setting experienced a transition from a 3- to a 2-bars system, which lasted for 69% of the total time. Recordings from Terracina, spanning 11.3 years, evidenced a complex variability of the bar structure, with continuous changes from a single- to a double-barred system, with a dominance of the latter configuration (74% of the total time). The beach in Sabaudia constantly exhibits two bars and a SPAW.

The change in the number of bars also shows site-specific characteristics. In Senigallia (first panel in Figure 4), the outer bar (b4) lost its visibility for almost a year, in fact changing the seabed bar configuration into a 3-bar system. The restoration of the 4-bar setting is due to the generation of new inner bar (b5) in February 2018. At Torre del Cerrano (second panel in Figure 4) the decrease of the bar number is coeval with the Senigallia stage transition (after February 2017) but was not followed by the generation of a new inner bar at the shoreline. In Terracina (third panel in Figure 4) the change in the bar number is always linked to the generation of a new bar near the shoreline (b2 in 2009, b3 in 2015, b4 in 2018).

The temporal distribution of distances between shoreline and barlines (Figure 5) highlights the presence of broader or thinner multiple peaks, connected to a more or less wide breaking region in the cross-shore direction. In the Adriatic sites (Senigallia and Torre del Cerrano; top panels in Figure 5), the distances of the inner and middle bars from the shoreline show a reduced dispersion around the respective peaks, thus, giving an indication of their linearity and relative stability in time. The distribution of inner bar distances for the Tyrrhenian sites (Sabaudia and Terracina; bottom panels in Figure 5), conversely, shows high dispersion around the peaks, suggesting more complex inner bar planar shapes and increased mobility of the bars in response to wave forcings.

Notably, at the Senigallia beach the distance from the shoreline of the inner and intermediate bars remain approximately constant in time, although an established pattern of generation and offshore migration of the bars occurs, in accordance with the NOM model [15,19]. The inner bars b1 (before February 2018) and b5 (after February 2018) show a position peak at around 40 m from the shoreline; in the same way, the first intermediate bars b2 (before February 2018) and b1 (after February 2018) have a position peak of about 80 m from the shore.

The seabed morphology observed at Terracina is representative of a 10-years nearshore morphological evolution after the artificial nourishment of the beach. The reshaping of this area is characterized by a variable “equilibrium” condition of the sandbars that follows a decrease of the surf zone size. Since the distance reached by most offshore bar is never reached by the other barlines afterwards, the distribution of the shoreline-bar distances well shows a decrease of the distance of inner and outer bars in time (Figure 5).

The distribution of shoreline-barlines distances at Sabaudia differs from those observed in Senigallia and Terracina since, even if the outer bar (b2) experiences the largest range in positioning (from 150 m to 350 m from the shoreline), no new bar is generated at the shoreline in response; transitional forms (SPAW) evolve instead, at first between the outer and inner bar, and then between the inner bar and the shoreline (Figure 5), suggesting a very complex morphology evolution both of the foreshore and in the outer surf zone. Similar behaviours are recorded for crescentic bars located along other Mediterranean sites [56].

The Torre del Cerrano bar setting, finally, exhibits a rather constant position of the inner and intermediate bars, located respectively at around 45–50 m and 100–150 m from the shoreline (Figure 5).

### 3.3. Rates of Variability

The net shoreline displacement rates (end point rates, EPR) do not display specific trends, apart from the case of Terracina, where a 40 m shoreline retreat is registered over 11 years (Table 6).

The monthly rates of shoreline displacements (Figure 6) show that the highest rates of displacement were experienced at the change in bar setting, reaching their maximum values in Terracina and Senigallia. On the opposite, the results for Sabaudia and Torre del Cerrano are always within the range of uncertainty.

Finally, at all sites the maximum rate of cross-shore variability reached its highest value at the outermost bars (b3 and b4 for Senigallia, b3 for Torre del Cerrano, b1 for Terracina, and b2 for Sabaudia; Figure 7) independently of the temporal windows used for the mean rate calculation (7 days, 15 days or 1 month). Furthermore, the rates are almost twice as large for the beaches along the Tyrrhenian sea than for those along the Adriatic sea.

The highest rate of cross-shore movement was recorded by the SPAW in Sabaudia, also regardless of the temporal windows used for the rate computation.

## 4. Discussion and Conclusions

The above findings provide a preliminary inter-site comparison of rates, amplitudes and modes of morphologic variability for the analyzed beach systems, with a focus on the long-term evolution. For all the sites, the rates of variability of sandbar cross-shore positions are much larger than for the shoreline, providing further indication on the large relative importance of the nearshore morphologic variability on the overall beach evolution (e.g., see [57]).

The beaches of Terracina and Senigallia experienced a similar net offshore bar migration process, but with some differences in rates, likely related to differences in the Iribarren number (much larger for the Tyrrhenian sites than for the Adriatic sites). During the observation period in Terracina (11 years), the generation of three bars was related to the fast seaward migration of the outermost bar. The width of the emerged beach sensibly decreased at the times in which the new inner bars are generated (this is particularly evident in 2009; see Figure 4), but remained stable afterwards. The evidence of outer bar decay was always related to waves breaking on a wider area (indicating a flatter bar) until the intermediate bar was forced to weld to the decaying outer bar. In Senigallia the nearshore setting experienced a more regular evolution, with lower but progressive offshore movements of the whole bar system, even if the two most offshore bars experienced higher migration rates. Shoreline displacement reached its maximum values during the near-shoreline generation of the inner bar, after which it remained fairly stable.

During the 2.9 years of observation of the Torre del Cerrano beach, no specific trends of sandbar migration were recorded, even though during the latest period of observation a fast offshore movement of the outer bar was recorded. In this site the shoreline is progressively advancing. The environmental parameters are similar to those of Senigallia, where a clear NOM-type bar evolution pattern occurs. Such difference in behaviour seems to be related to the largest wave obliquity observed in Senigallia, this leading to larger longshore currents and associated larger seabed friction, giving larger sediment mobility.

The Sabaudia site is mainly characterized by complex foreshore morphologies and very high rates of outer bars seaward motion. No erosional trends of the shoreline are observed, but alongshore moving transient features indicate a three-dimensional pattern of variability along this analyzed portion of beach, where the most of high waves approached the coast with small angles.

This preliminary inter-site comparison suggests some findings that seem to agree with similar studies on the matter:the sandbar morphologic variability is much larger than the shoreline displacement. This is very likely related to the exposure of sandbars to more intense forcing flows than those experienced by the near-shoreline sediments;for the first time, evidence of NOM-type sand bar evolution patterns along Italian coasts was recorded;migration rates of the sandbars during NOM seem to be significantly influenced by the wave approach angle, as well as the nearshore slope (see also [58,59,60,61,62,63,64]), which in turn translates into different values of the Iribarren number;the major erosional phenomena along the analyzed beaches coincide with discrete changes in the nearshore morphological patterns that occur in relation to the rapid seaward migration of the outer bars or to the decay of the offshore bar. In view of the results of our analysis, the reciprocal roles of wave obliquity and beach slope can be summarized as per Table 7;the larger shoreline displacements have been observed along those beaches that are characterized by a NOM-type evolution of the entire sandbar system. This suggests that quasi-steady sandbar systems are related to more stable shorelines. However, more detailed analyses are needed to verify if such behaviour also leads to more resilient coastlines (i.e., with a lesser retreat in the long term);high-frequency remote sensing technology can provide fundamental insight about such long-term shoreline evolution. Moreover, it can provide important information for coastal management purposes, especially along natural coasts were dynamics could be very fast and where no other (historic) data are available.

## Figures and Tables

**Figure 1 sensors-19-01854-f001:**
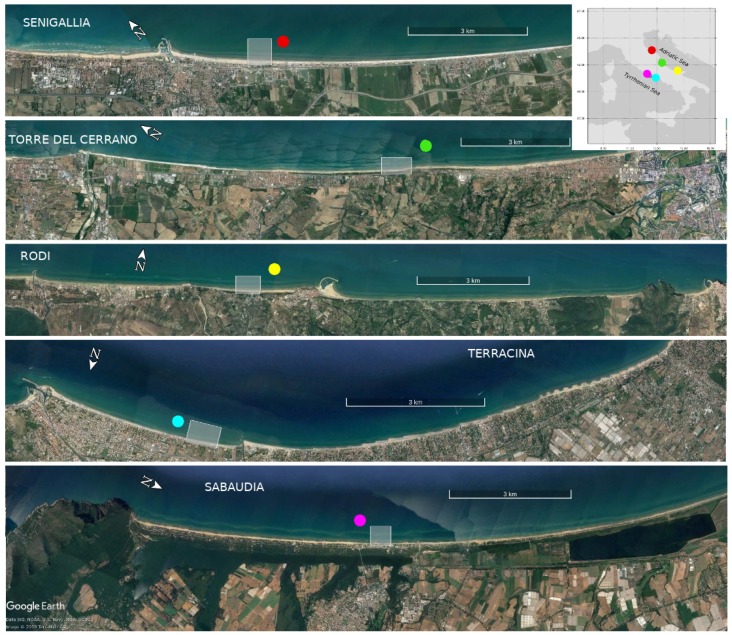
The five investigated locations (pictures taken from Google Earth). From top to bottom: Senigallia, Torre del Cerrano, Rodi Garganico, Terracina and Sabaudia. The shaded areas indicate the video-monitored portions of beach.

**Figure 2 sensors-19-01854-f002:**
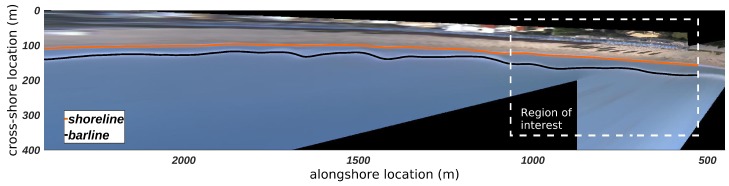
An example of rectified and georeferenced Timex image from the video-monitoring station in Terracina. The remotely sensed locations of the shoreline and the bar are shown with the orange and the black line, respectively. The dashed line box shows the region of interest along the Terracina coast.

**Figure 3 sensors-19-01854-f003:**
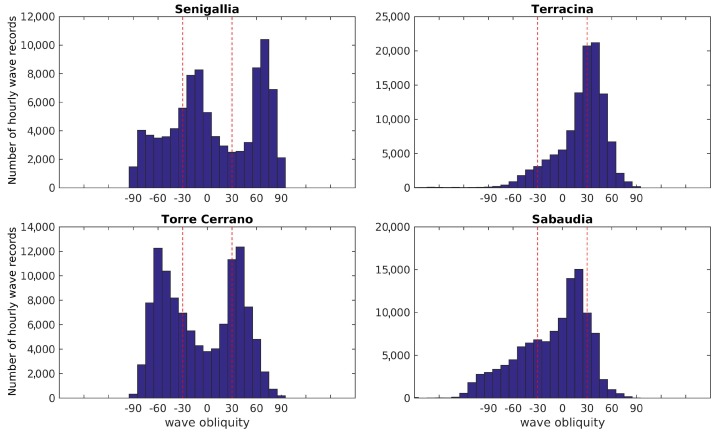
Distribution of wave obliquity at Senigallia, Terracina, Torre del Cerrano and Sabaudia. The red dashed lines demarcate the $\pm 30^{\circ }$ interval of wave incidence with respect to the shore normal. In the *y*-axis the number of hourly wave records coming from a given direction interval throughout the total observation period is given.

**Figure 4 sensors-19-01854-f004:**
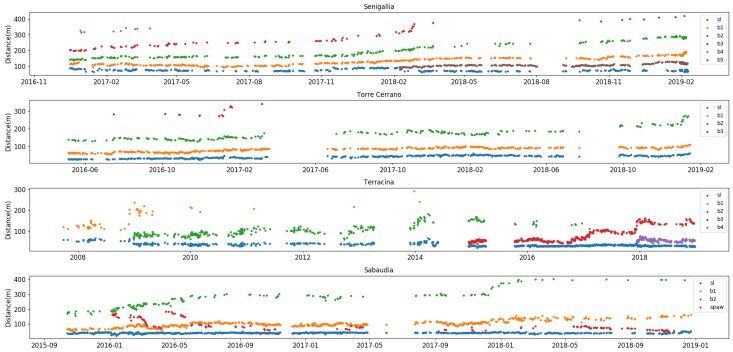
Time evolution of locations of shoreline and barlines in Senigallia, Torre del Cerrano, Terracina and Sabaudia. Each dot represents the daily-averaged location of the relative beach feature.

**Figure 5 sensors-19-01854-f005:**
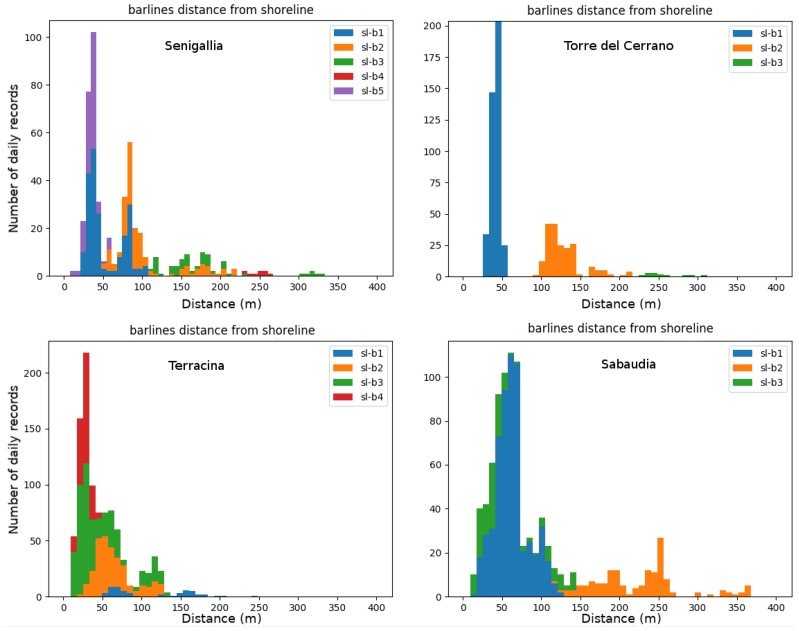
Distribution of distances between shoreline and bars at Senigallia, Torre del Cerrano, Terracina and Sabaudia. In the *y*-axis the number of daily occurrences in which the distance between a bar and the shoreline could be evaluated is reported.

**Figure 6 sensors-19-01854-f006:**
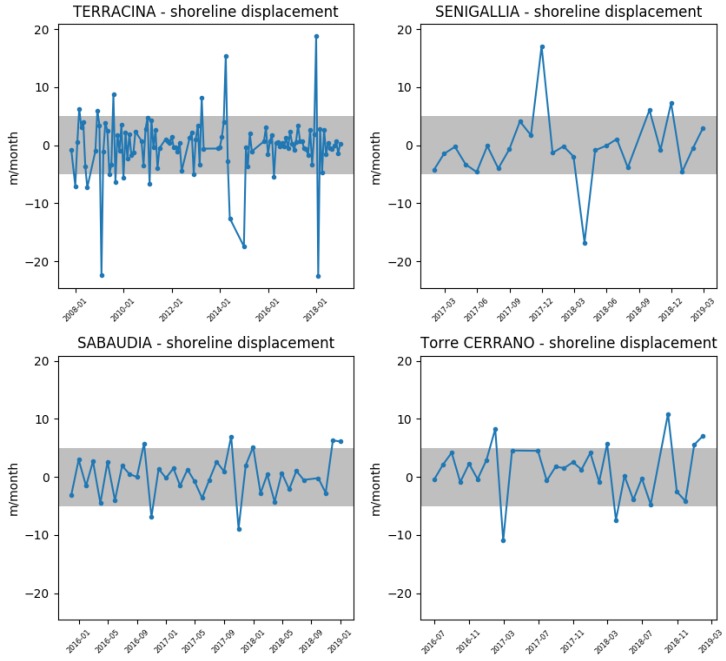
Maximum shoreline displacement rates at Senigallia, Torre del Cerrano, Terracina and Sabaudia, calculated from monthly-resampled time series of shoreline displacement. The shaded area indicates a displacement rate range from −5 to +5 m/month. Negative values correspond to shoreline retreat, while positive values show shoreline advancement.

**Figure 7 sensors-19-01854-f007:**
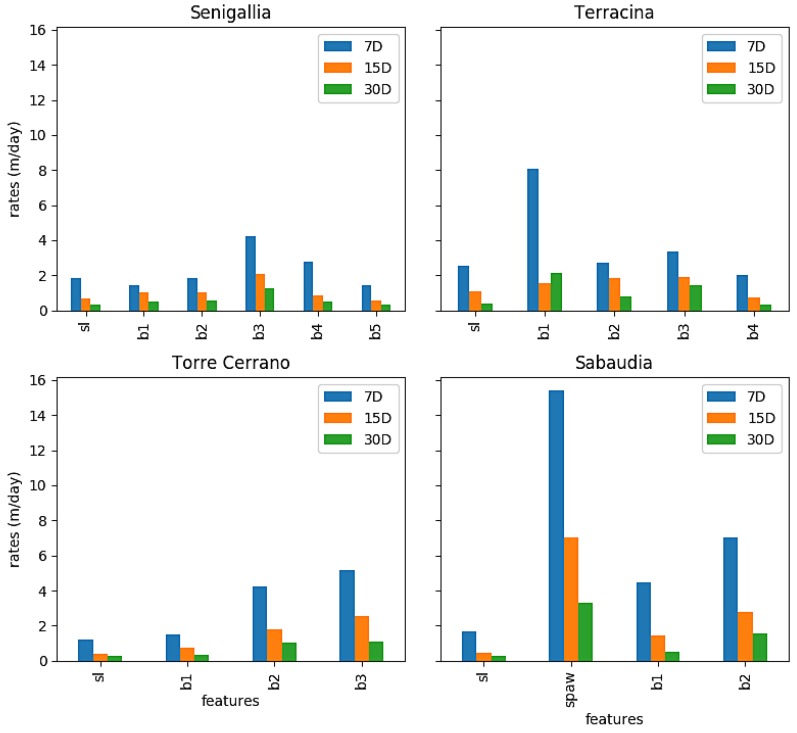
Maximum rates of variability for sampled morphology features at the investigated sites. The rates are computed with different temporal windows: 7 days (blue bars), 15 days (orange bars), and 1 month (green bars).

**Table 1 sensors-19-01854-t001:** Specifics of the deployed video-monitoring stations.

Location	Sea	Coordinates	Height (m)	Cameras Azimuth (${}^{\circ }$N)	Focal Length (mm)	Monitored Beach (m)
Senigallia	Adriatic	$43^{\circ }\;42^{\prime }\;24^{\prime \prime }\;N$ $13^{\circ }\;14^{\prime }\;14^{\prime \prime }\;E$	28.0	60${}^{\circ }$ 115${}^{\circ }$	8 12	800
Torre del Cerrano	Adriatic	$42^{\circ }\;35^{\prime }\;06^{\prime \prime }\;N$ $14^{\circ }\;05^{\prime }\;21^{\prime \prime }\;E$	37.5	40${}^{\circ }$ 330${}^{\circ }$	12 25	1000
Rodi Garganico	Adriatic	$41^{\circ }\;55^{\prime }\;39^{\prime \prime }\;N$ $15^{\circ }\;52^{\prime }\;34^{\prime \prime }\;E$	54.0	310${}^{\circ }$ 345${}^{\circ }$	8 8	500
Sabaudia	Tyrrhenian	$41^{\circ }\;18^{\prime }\;00^{\prime \prime }\;N$ $13^{\circ }\;00^{\prime }\;33^{\prime \prime }\;E$	21.1	165${}^{\circ }$ 240${}^{\circ }$	12 6	800
Terracina	Tyrrhenian	$41^{\circ }\;17^{\prime }\;03^{\prime \prime }\;N$ $13^{\circ }\;12^{\prime }\;27^{\prime \prime }\;E$	39.0	95${}^{\circ }$ 165${}^{\circ }$	25 8	1000

**Table 2 sensors-19-01854-t002:** The image datasets used for the study. For each location, the observational time intervals and the total number of observed days are reported. For each feature detected at a specific location, the number of days that the feature has been detected and its percentage of occurrence over the total number of days are also given.

Location	Dataset Specifics & Days of Feature Occurrence					
**Senigallia**	Time interval: 16 December 2016–9 February 2019 Number of days: 786					
feature	sl	b1	b2	b3	b4	b5
n° of days	370	447	282	89	9	307
% occurrence	47	57	36	11	19	39
**Torre del Cerrano**	Time interval: 15 May 2016–14 January 2019 Number of days: 985					
feature	sl	b1	b2	b3		
n° of days	515	458	214	15		
% occurrence	52	47	22	2		
**Sabaudia**	Time interval: 13 October 2015–22 December 2018 Number of days: 1167					
feature	sl	spaw	b1	b2		
n° of days	812	155	604	157		
% occurrence	70	13	52	13		
**Terracina**	Time interval: 28 September 2007–28 December 2018 Number of days: 4110					
feature	sl	b1	b2	b3	b4	
n° of days	1307	54	285	484	218	
% occurrence	32	1	7	12	5	

**Table 3 sensors-19-01854-t003:** Wave climate statistics for each study site: wave height threshold for storm classification $H_{s,0.97}$, mean significant wave height at classified storm peaks $\langle H_{s,peak}\rangle$, and mean peak period at classified storm peaks $\langle T_{p,peak}\rangle$.

	Senigallia	Torre del Cerrano	Rodi Garganico	Terracina	Sabaudia
$H_{s,0.97}$ (m)	2.04	1.91	1.99	2.26	2.41
$\langle H_{s,peak}\rangle$ (m)	2.81	2.67	2.67	3.18	3.33
$\langle T_{p,peak}\rangle$ (s)	7.30	7.42	7.41	8.71	8.59

**Table 4 sensors-19-01854-t004:** Environmental and classification parameters used to characterize the main morphological behaviour of the study sites.

	Slope (%)	TR (m)	$d_{50}$ (mm)	Ω [13]	$B^{*}$ [49]	RTR [14]	Ir [50]
Senigallia	0.5	0.6	0.25	11.3	497	0.2	0.03
Torre del Cerrano	0.5	0.5	0.30	8.6	447	0.2	0.03
Rodi Garganico	0.8	0.4	0.30	8.6	209	0.1	0.05
Sabaudia	1.3	0.4	0.30	9.3	55	0.1	0.09
Terracina	1.7	0.4	0.30	8.9	67	0.1	0.08

**Table 5 sensors-19-01854-t005:** Persistence of sand bars settings at the study sites. The left section of the table shows the percentages of the observation period in which a specific bar number has been observed. The right section of the table shows the number of consecutive days in which a specific bar configuration (given under each number) has been observed.

Location	Number of Bars				Consecutive Days of Bar Setting					
	Single	Double	Three	Four						
Senigallia			40%	60%	102 four	316 three	368 four			
Torre del Cerrano		69%	31%		307 three	678 double				
Terracina	26%	74%			455 single	1863 double	317 single	795 double	303 single	377 double
Sabaudia		100%			all double					

**Table 6 sensors-19-01854-t006:** Net shoreline displacement rates (as end point rates) at Senigallia, Torre del Cerrano, Terracina and Sabaudia.

	Senigallia	Torre del Cerrano	Terracina	Sabaudia
EPR (m/month)	−0.5	0.9	−0.2	0.3

**Table 7 sensors-19-01854-t007:** Schematic of the roles of beach slope and wave obliquity on the bars-shoreline system evolution.

	Low Wave Obliquity	High Wave Obliquity
**Low Ir**	No trend–stable bars and shoreline	Cyclic behaviour (NOM) with low rates of change
**High Ir**	3D patterns with high rates of change	Cyclic behaviour (NOM) with high rates of change

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
