# Peer review of "Monitoring for Coastal Resilience: Preliminary Data from Five Italian Sandy Beaches"

_sensors, 2019, doi:10.3390/s19081854_

Round 1

Reviewer 1 Report

The reference could be updated.

looking at video monitoring systems, many papers have been produced in the last years:

i.e.

Valentini, N., Saponieri, A., Damiani, L. (2017). A new video monitoring system in support of Coastal Zone Management at Apulia Region, Italy. J. Ocean & coastal management142, 122-135

Ibaceta, R., Almar, R., Catalán, P., Blenkinsopp, C., Almeida, L.,  Cienfuegos, R. (2018). Assessing the Performance of a Low-Cost Method for Video-Monitoring the Water Surface and Bed Level in the Swash Zone of Natural Beaches. J. Remote Sensing10(1), 49

Bonaldo, D.; Antonioli, F.; Archetti, R.; Bezzi, A.; Correggiari, A.; Davolio, S.; de Falco, G.; Fantini, M.; Fontolan, G.; Furlani, S.; Gaeta, M. G.; Leoni, G.; Lo Presti, V.; Mastronuzzi, G.; Pillon, S.; Ricchi, A.; Stocchi, P.; Samaras, A. G.; Scicchitano, G.; Carniel, S. (2019). Integrating multidisciplinary instruments for assessing coastal vulnerability to erosion and sea level rise: lessons and challenges from the Adriatic Sea, Italy.  J. Coastal Conservation, pp.1-19, vol. 23(1)

As stated in row 15, "capability of the beach to dissipate and absorb the energy of waves" is relevant. By this way it could be mentioned the oportunity to measure runup (related to energy content) on the beach by using video monitoring systems.

Some more details could be provided on the videomonitoring system. 

i.e.: the AA chosen "only one single image per day", probaly it could be better to chose the daily mean of all available immages, unless the processing time is very long.

About the descriptions of sites, it could be interesting to report the presence of works influencing morphodynamics (i.e. port of Rodi)

In row 181 AA state that wave approach the coast normally at Rodi Garaganico. It is suggested to verify. In other papers NW wave direction is assumed.

The bar eight is not considered among the parameters influencing beach morphodynamics: even if the monitoring system doesn't provide any data, this parameter could be mentioned.

Author Response

REPLY LETTER FOR REVIEWER #1

We thank the Reviewer for his/her interest in our paper and for his/her useful criticism. We have taken into due account all the Reviewer’s comments and performed a thorough revision that satisfies virtually all the requests made by the Reviewer. An item-by-item reply follows, with the order given in the Reviewers' review. For ease of inspection, the new text is implemented in red in the revised manuscript.

The reference could be updated.

looking at video monitoring systems, many papers have been produced in the last years: i.e.

Valentini, N., Saponieri, A., Damiani, L. (2017). A new video monitoring system in support of Coastal Zone Management at Apulia Region, Italy. J. Ocean & coastal management, 142, 122-135

Ibaceta, R., Almar, R., Catalán, P., Blenkinsopp, C., Almeida, L., Cienfuegos, R. (2018). Assessing the Performance of a Low-Cost Method for Video-Monitoring the Water Surface and Bed Level in the Swash Zone of Natural Beaches. J. Remote Sensing, 10(1), 49

Bonaldo, D.; Antonioli, F.; Archetti, R.; Bezzi, A.; Correggiari, A.; Davolio, S.; de Falco, G.; Fantini, M.; Fontolan, G.; Furlani, S.; Gaeta, M. G.; Leoni, G.; Lo Presti, V.; Mastronuzzi, G.; Pillon, S.; Ricchi, A.; Stocchi, P.; Samaras, A. G.; Scicchitano, G.; Carniel, S. (2019). Integrating multidisciplinary instruments for assessing coastal vulnerability to erosion and sea level rise: lessons and challenges from the Adriatic Sea, Italy. J. Coastal Conservation, pp.1-19, vol. 23(1)

Thank you for the useful comment. Following your suggestion, the bibliography has been complemented with the mentioned works. The first work is cited as an example of video-monitoring station deployed along Italian coasts (L59). The second manuscript is cited as an example of the use of video-monitoring for swash zone observation purposes (L57). The third manuscript is cited to support the need for more effective and informed strategies for coastal management (L61).

As stated in row 15, "capability of the beach to dissipate and absorb the energy of waves" is relevant. By this way it could be mentioned the opportunity to measure runup (related to energy content) on the beach by using video monitoring systems.

In response to this observation, we have revised the introductory part (L48-61) to better express the usefulness of video-monitoring systems, and to include a few applications.

Some more details could be provided on the videomonitoring system.
i.e.: the AA chosen "only one single image per day", probaly it could be better to chose the daily mean of all available immages, unless the processing time is very long.

In view of the time scales of the dynamics of interest here (months to years) the choice of the specific daily image in use has little bearing. The Reviewer’s suggestion is very appropriate in the case of dynamics with daily time scales. Besides, the choice of daily-averaged images is more significant for environments with larger tide excursions.

About the descriptions of sites, it could be interesting to report the presence of works influencing morphodynamics (i.e. port of Rodi)

Following this suggestion, we have included a mention to the nearby port of Rodi Garganico in the description of the study site (L94-96).

In row 181 AA state that wave approach the coast normally at Rodi Garaganico. It is suggested to verify. In other papers NW wave direction is assumed.

We have included also the NW direction among the wave directions for Rodi Garganico (L201).

The bar eight is not considered among the parameters influencing beach morphodynamics: even if the monitoring system doesn't provide any data, this parameter could be mentioned.

We have mentioned this in L156-160. In the new text we have also specified that some studies are being undertaken on how to estimate the bathymetry from video-monitoring images (L160).

We thank again the Reviewer.

p { margin-bottom: 6.25px; direction: ltr; line-height: 115%; text-align: left; }

Reviewer 2 Report

Paper is very interesting and gives a lot of materials for the new hypotheses and will stimulate the further investigations.

Some improvements I suggest below.

9 Acronym NOM is not very popular, the explanations exists in the text of paper only, please or explain it in the abstract or avoid.

132 It should be clarified what is BLIM – algorithm or program or some formula?

What are the errors of estimating of sand bar crest position by maximum of brightness as discussed in Van Enckevort, I.M.J., Ruessink, B.G., 2001 Effect of hydrodynamics and bathymetry on video estimates of nearshore sandbar position. Journal of Geophysical Research 106, 16969–16980?

153 It should be clarified that is n – is it grow with the distance from the shore or with the temporal occurrence.

In Table 2 it is unclear what is “count” - number of days? Please explain in the text.

What is “count” at Figure 3 – number of waves?

199 Please explain clearly how you take into account the wave shoaling

207 Values of what?

225 I can’t understand what «the way sites» is, but may be my English is not enough

255 Please write how the NOM model [16,32] explain the bar position behavior. What is the NOM model? Previously the NOM behavior only was mentioned.

290-293 It is unclear who make this conclusion – the authors of the paper or the authors of [46]

324-325 May be more correct will be: depend on wave approach angle instead of beach orientation? Does it confirm or contradict to results of [4753]?

What is “count” at Figure 5?

Author Response

REPLY LETTER FOR REVIEWER #2

We thank the Reviewer for his/her interest in our paper and for his/her useful criticism. We have taken into due account all the Reviewer’s comments and performed a thorough revision that satisfies virtually all the requests made by the Reviewer. An item-by-item reply follows, with the order given in the Reviewers' review. For ease of inspection, the new text is implemented in red in the revised manuscript.

9 Acronym NOM is not very popular, the explanations exists in the text of paper only, please or explain it in the abstract or avoid.

We believe that an explanation of the NOM model is more appropriate within the Introduction rather than in the Abstract, since it would make the abstract much longer than necessary. To comply with the Reviewer’s observation, we have expanded the acronym NOM in the Abstract to make its meaning explicit (L9), and further expanded the explanation of the NOM model in the Introduction (L35-40).

132 It should be clarified what is BLIM – algorithm or program or some formula?

BLIM is an algorithm, as we have already stated in the sentence “In this algorithm, pixels within a region of interest are scanned along the image in order to detect the position lines of maximum pixel intensity values” (L148-149).

What are the errors of estimating of sand bar crest position by maximum of brightness as discussed in Van Enckevort, I.M.J., Ruessink, B.G., 2001 Effect of hydrodynamics and bathymetry on video estimates of nearshore sandbar position. Journal of Geophysical Research 106, 16969–16980?

For this work we could not use the method by Van Enckevort & Ruessink (2001), which relies on available bathymetries. We had no such bathymetries for all sites, hence we used a different approach. We used different cut-off thresholds of position as function of the feature at hand and increasing from the shore (±5 m) to the offshore (±20 m). All consecutive feature motions smaller than such thresholds were regarded as fictitious displacements related to hydrodynamic and morphologic forcing, as described in van Enckevort and Ruessink (2001).

To further avoid the risk of identifying fictitious movements due to hydrodynamic causes, all the evaluations of displacement rates of the sampled morphologies were made on sub-sampled datasets (7, 15 and 30 days). An improvement of the spatial resolution of extracted morphologies for the study sites will require specific bathymetric surveys, in order to increase the spatial resolution and then the capability to resolve storm-induced morphologic variability in the short time.

153 It should be clarified that is n – is it grow with the distance from the shore or with the temporal occurrence.

n increases with both the offshore position of the bars and the temporal occurrence. b1 is always the bar that is closest to the shoreline at the beginning of the observation period. b2 is the second-closest to the shoreline at the beginning of the observation period. b3 is the third-closest… and so on, to cover all the m bars that are visible at the start of the observation period. As other bars appear later on, they are given numbers m+1, m+2... and so on.

In Table 2 it is unclear what is “count” - number of days? Please explain in the text.

count” is the number of days in which the specific bathymetric features occurred over the whole observation period. We have replaced the ambiguous word “count” with “n° of days” in Table 2 and explained it also in the text (L177-178).

What is “count” at Figure 3 – number of waves?

Count in Figure 3 stands for the number of hourly storm wave records coming from a specific direction range, as specified in the figure caption.

199 Please explain clearly how you take into account the wave shoaling

Following this suggestion, we have explained how wave shoaling has been accounted for in the evaluation of the breaking wave height (L219-222)

207 Values of what?

The sentence is referred to “values of Ω”. This has been clarified in L228.

225 I can’t understand what «the way sites» is, but may be my English is not enough

We have clarified the sentence (L246-247).

255 Please write how the NOM model [16,32] explain the bar position behavior. What is the NOM model? Previously the NOM behavior only was mentioned.

The NOM model has been briefly explained in the Introduction (L35-40), and we here observe that the bar system in Senigallia follows that model (bar generation, migration, and degradation) quite well.

290-293 It is unclear who make this conclusion – the authors of the paper or the authors of [46]

The statement “For all the sites, the rates of variability of sandbar cross-shore positions are much larger than for the shoreline, providing further indication on the large relative importance of the nearshore morphologic variability on the overall beach evolution” is made by the Authors. We cite the manuscript as further confirmation of how the beach variability is mainly dominated by the interannual patterns of bar formation, migration, and decay.  

324-325 May be more correct will be: depend on wave approach angle instead of beach orientation? Does it confirm or contradict to results of [47–53]?

We agree with this observation: the beach orientation is clearly connected with how the waves approach the land, but certainly the wave angle is an important factor on the migration rates. We have corrected the sentence accordingly (L346-348).

What is “count” at Figure 5?

Count in Figure 5 stands for the number of daily images in which a distance between the bars and the shoreline could be evaluated. We better clarified this aspect in the figure caption.

We thank again the Reviewer for the suggestions given.

p { margin-bottom: 6.25px; direction: ltr; line-height: 115%; text-align: left; }p.western { font-size: 15px; }p.cjk { font-size: 15px; }

Reviewer 3 Report

Congrats for the work. No comments regarding its scientific and technical content, not the structure and presentation. The only concern, if any, goes for the English. Please go back over the paper and check some spelling and structure, specially in the Intro section. Try to shorten phrases in order to provide with the same clear ideas but with few words.

Author Response

REPLY LETTER FOR REVIEWER #3

 We thank the Reviewer for his/her kind appreciation of our work. As suggested, we have revised the English style and spelling of the manuscript, with particular attention to the introductory part. The sentences that have been shortened, rephrased or somewhat changed have been evidenced in red in the revised manuscript.

We thank again the Reviewer.

p { margin-bottom: 6.25px; direction: ltr; line-height: 115%; text-align: left; }